# LinearizeLLM: An Agent-Based Framework for LLM-Driven Exact Linear Reformulation of Nonlinear Optimization Problems

## Abstract

Reformulating nonlinear optimization problems is largely manual and expertise-intensive, yet it remains essential for solving such problems with linear optimization solvers or applying special-purpose algorithms. We introduce *LinearizeLLM*, an agent-based framework that solves this task by leveraging Large Language Models (LLMs). The framework assigns each nonlinear pattern to a *reformulation agent* that is explicitly instructed to derive an exact linear reformulation for its nonlinearity pattern, for instance, absolute-value terms or bilinear products of decision variables. The agents then coordinate to assemble a solver-ready linear model equivalent to the original problem. To benchmark the approach, we create a dataset of 20 real-world nonlinear optimization problems derived from the established ComplexOR dataset of linear optimization problems. We evaluate our approach with several LLMs. Our results indicate that specialized LLM agents can automate linearization tasks, opening a path toward fully conversational modeling pipelines for nonlinear optimization.

## 1 Introduction

Complex real-world optimization problems often involve nonlinear relationships between decision variables that make optimization problems computationally challenging to solve. A well-established strategy in Operations Research (OR) is to derive approximations or relaxations of nonlinear optimization problems (NLPs), transforming them into more tractable optimization problems (Mohagheghi & Rebennack, 2015). These approximations allow solvers such as Gurobi 12 (Gurobi Optimization, LLC, 2024) to leverage their powerful solution techniques on problem types, such as Linear Optimization Problems (LPs) or Mixed Integer Linear Optimization Problems (MILPs). In practice, however, designing effective reformulations of NLPs typically requires expert knowledge. Practitioners without OR expertise often struggle with intractable models, because they are unaware of the sophisticated reformulation techniques. This gap between advanced OR theory and what non-experts can readily apply represents a significant barrier to the broader use of optimization in industry and science (Chen et al., 2023).

Meanwhile, the rise of Large Language Models (LLMs) offers a compelling opportunity to bridge this expertise gap (Wasserkrug et al., 2025). Recent LLM-based approaches have demonstrated the ability to perform complex reasoning tasks in OR. (Xiao et al., 2024) have begun to position LLMs as high-level translators of textual problem descriptions into formal optimization problems and even solve them with Chain-of-Experts. Additionally, OptiMUS introduced by (AhmadiTeshnizi et al., 2024) is an LLM-driven framework that can formulate an MILP from a textual description, write corresponding solver code, and iteratively refine the optimization problem based on solution feedback. These systems have already shown that LLMs can alleviate the heavy dependence on domain experts in the modeling phase, opening the door for non-experts to harness optimization technology. However, an equally critical challenge remains under-explored: Once an initial complicated NLP is formulated from language, how can we reformulate that optimization problem into an equivalent linear model? In other words, beyond producing an optimization problem from language through LLM-based systems can an LLM-based system also improve the model's tractability by linearizing NLPs?

In this study we focus on nonlinear patterns that admit exact linear reformulations, such as $|\cdot|$, $\min/\max$, binary–continuous products, linear fractionals or monotone transformations, while noting that relaxation-based techniques – e.g., McCormick envelopes for purely continuous bilinears (McCormick, 1976) – are also compatible with our agent architecture. This question is vitally important because even an algebraically correct formulated optimization problem may be impractical to be solved if it involves nonlinear functions. Unfortunately, too many practitioners lack access to this repertoire of techniques, and current LLM systems do not yet fill this void. Recent evaluations highlight this gap (Wasserkrug et al., 2025): When asked to reformulate a given optimization problem, an LLM often fails to produce an equivalent (or "nearly" equivalent) optimization problem.

This paper introduces *LinearizeLLM*, a fully LLM-driven framework that automatically converts a broad class of NLPs into algebraically equivalent linear models. Specifically, as our main contributions, we

    (i) propose an agent architecture in which each nonlinear term is handled by a specialized *reformulation agent* instructed to derive an exact linearization pattern after reading in the original problem formulation in LaTeX code; the agents then coordinate to assemble a solver-ready model;

  (ii) release a benchmark of 20 real-world instances, obtained by injecting exactly linearizable nonlinearity patterns into the publicly available ComplexOR dataset (Xiao et al., 2024);

 (iii) conduct an empirical evaluation with *Gemini 2.5 Flash* and OpenAI's *o3*, including head-to-head baselines, a one-shot comparison, and a context-blind ablation study, showing that our agent framework consistently outperforms strong baselines and achieves high exact linearization success rates.

## 2 MIXED-INTEGER NONLINEAR PROGRAMMING

Mixed-Integer NonLinear Problems (MINLPs) arise in a wide range of real-world applications, including process systems engineering, energy operations, logistics, and finance, where discrete decisions must be taken in the presence of nonlinear dynamics (Belotti et al., 2013).

Formally, a general MINLP can be written as:

$$\min_{x\in\mathbb{R}^n,\,y\in\mathbb{Z}^m} \quad f(x,y)$$
$$\text{s.t.} \quad g_i(x,y) = 0, \quad \forall i$$
$$h_j(x,y) \leq 0, \quad \forall j$$

where $f(x,y)$, $g_i(x,y)$ and $h_j(x,y)$ are potentially nonlinear functions in $(x,y)$, and $y$ contains integer-valued decision variables. If all functions are linear, the problem is a MILP; if, in addition, there are no $y$ variables, i.e., $m=0$, then the problem is an LP.

## 3 MOTIVATION

Modern MILP solvers incorporate presolve routines and even automatic detection of certain nonlinear patterns. However, these built-in features are limited and opaque. By contrast, an explicit reformulation handled outside the solver provides transparency and auditability. The user (or model auditor) can inspect the introduced auxiliary variables and linear constraints, verifying that they correctly represent the original nonlinear relations. Such auditability builds trust: the reformulated model is human-readable and can be double-checked line by line, unlike solver-internal transformations that are hidden from view. Auditability is crucial in high-stakes applications where one must ensure the reformulation has not altered the problem's intent or feasibility region.

Second, portability and integration are crucial aspects when reformulating NLPs into LPs or MILPs. Once the original model is recast as a pure LP/MILP, it can be solved by any state-of-the-art LP/MILP solver. Portability also means the same LP/MILP file can be handed off unchanged to commercial solvers such as Gurobi and CPLEX, open-source engines like HiGHS (Huangfu et al., 2023) and CBC (Forrest & Ralphs, 2022), or even cloud-hosted MILP services such as Google's

Cloud Optimization API (Google LLC, 2025), without tying the user to any particular nonlinear solver or license regime.

Third, an explicit reformulation allows for systematic coverage of nonlinear patterns. Solvers might linearize simple cases (e.g., rewriting a simple on/off constraint such as $y \leq Mz$, with $z$ as a binary), but they cannot handle every situation (Vielma, 2015). In contrast, a comprehensive reformulation approach can be designed to catch a wide range of nonlinear patterns.

Therefore, we introduce *LinearizeLLM* which is an LLM agent-based reformulation framework that harnesses LLMs to deliver exact, solver-ready LP/MILP counterparts of NLPs. *LinearizeLLM* (i) generates a fully documented set of auxiliary variables and constraints, giving auditors line-by-line traceability; (ii) outputs its model in standard algebraic form, so the same file can be fed to any LP/MILP engine or embedded as a linear sub-problem inside decomposition schemes; and (iii) employs pattern-specialized LLM agents that systematically recognize and linearize nonlinear patterns beyond the cases handled by solver presolve. In doing so, *LinearizeLLM* transforms algebraic expertise into an automated, transparent, and portable service layer, unlocking the power of modern LP/MILP technology for practitioners who would otherwise be constrained by the limits of native nonlinear optimization.

## 4 RELATED WORK

### 4.1 LLMS FOR MODEL FORMULATION

Early work on natural language to optimization focused on the modeling task, i.e., formulating MILPs from problems described with natural language. (Ramamonjison et al., 2022) introduced NL4Opt, a publicly available dataset and NeurIPS 2022 competition with the task of translating real-world problems into LPs. Several other studies later focused on this problem. (Li et al., 2023) developed a three-phase framework to formulate MILPs from natural text and extended the N4LOpt dataset to evaluate their approach. Additional work on this task includes OptiMUS, introduced by (AhmadiTeshnizi et al., 2024), an LLM based agent, that models natural language problems as MILPs, writes and evaluates the solver-ready code. (Xiao et al., 2024) is closest to our work. The authors introduced a multi-agent framework *Chain-of-Experts*, where each agent is assigned to a specific task. Their framework is capable of generating solver-ready code for OR problems and is evaluated on the new *ComplexOR* dataset.

However, those approaches assume linearity of the underlying problem and nonlinear optimization problems are consequently not covered. In the OptimAI approach (Thind et al., 2025), a suitable solver is selected based on the structure and requirements of the given optimization problem. However, no reformulation of the problem is performed, and the problem is solved in its original mathematical form. A recent study by (Wasserkrug et al., 2025) tested ChatGPT's (OpenAI, 2023) ability to perform algebraic reformulations. When asked to replace a nonlinear absolute-value constraint with linear constraints (a common linearization task), ChatGPT correctly produced an equivalent formulation in convex cases. However, for non-convex cases that require integer auxiliary variables, the model's answers were usually incomplete–it tended to omit the necessary binary variables, yielding incorrect formulations. This suggests that LLMs are capable of recognizing certain reformulation patterns (indeed, ChatGPT *knew* that "max" constraints can be rewritten as linear inequalities) but may fail to enforce logical consistency (such as branching via binaries) unless explicitly guided.

This highlights the need for explicit guidance or agent-based decomposition when exact linearization is required. To fill that research gap, we introduce *LinearizeLLM*, a multi-agent pipeline that reformulates every nonlinear pattern individually and guarantees solver-ready linear models. The next section details its workflow.

## 5 LINEARIZELLM WORKFLOW

*LinearizeLLM* transforms an NLP into an LP/MILP in three succinct stages: (1) a *detection agent* scans the original LATEX code and reports each unique nonlinear pattern; (2) pattern-specific *reformulation agents* reformulate every detected nonlinear pattern with its tightest detected linearization;

(3) the process repeats until no nonlinear patterns can be reformulated by the *reformulation agents*, yielding an LP/MILP optimization problem.

## 5.1 DETECTION AGENT

Each *detection agent* is an LLM-based agent responsible for identifying specific patterns of non-linearities in a given optimization model. A nonlinearity pattern refers to a recurring algebraic structure that represents a known nonlinear functional form like $x_k \cdot x_p \, \forall k, p$ for $x_k, x_p \in \mathbb{R}$. The agent receives contextual information including decision variable definitions and parameter structures through in-context learning to enhance pattern recognition.

**Instruction injection.** The system prompt names the pattern of nonlinearity patterns (e.g. absolute-value terms) and demands pattern-level grouping.

**Pattern-scan reasoning.** Guided by the prompt, the LLM sweeps through the LaTeX model and reasons step-by-step to decide whether a match belongs to one of the nonlinearity patterns.

**Aggregation and abstraction.** Whenever repeated index symbols are detected, the agent abstracts them into a single nonlinearity pattern instead of enumerating each instance.

## 5.2 REFORMULATION AGENTS

Each *reformulation agent* is an LLM-based module responsible for translating a specific nonlinear pattern into an equivalent LP or MILP formulation. While each agent targets a nonlinear pattern (e.g. bilinearity), all agents follow a unified multi-step process that ensures correctness, tightness, and interpretability of the reformulation. Each agent receives enhanced contextual information including decision variable definitions, parameter interactions in the optimization problem, and concrete parameter values through in-context learning.

**Structured reasoning.** Each agent is prompted with a structured, forward-thinking checklist which is a prompting strategy that encourages the LLM to reason step-by-step about what reformulation options are available, in what order they should be considered, and under what conditions each one is valid (Yao et al., 2022). The agent evaluates candidate techniques, stops at the first method that is both applicable and exact, derives tight bounds (or the smallest valid constant $M$ in the case of Big-$M$ reformulations), and drafts the corresponding linear constraints to replace the original nonlinear pattern. A final reflection step is used to verify formulation tightness and count the number of auxiliary variables introduced.

**Concise summary.** If the agent's internal Chain-of-Thought (CoT) reasoning is long or verbose, it automatically compresses the rationale into a short summary that includes: the recognized nonlinear pattern, the chosen reformulation technique, any computed bounds, and the definitions of auxiliary variables. This is inspired by the CoT prompting approach, where language models explicitly generate reasoning traces to improve optimization problem-solving overall success rate (Wei et al., 2022).

**Self-comment and checks.** Inspired by the Solo Performance Prompting (SPP) method (Zhou et al., 2023), each agent appends a short self-verification note to its output. This includes checks like `exactness verified`, `no remaining nonlinear patterns`, or flags any unresolved modeling issues for downstream agents in the reformulation loop.

## 5.3 THE WORKFLOW OF LINEARIZELLM

Let $\mathcal{P}$ be the set of solvable nonlinear optimization problems. For any $p \in \mathcal{P}$ all nonlinear patterns are assumed to be mutually independent and not nested. The original problem instance provided by the user is denoted by $p^{(0)} \in \mathcal{P}$. We denote $\Theta$ as the set that contains all parameter sets $\theta_p$, $\forall p$ for optimization problem $p$. $\theta_p$ be the set that contains all parameters (e.g., index sets) associated with problem $p$. Each $\theta_p$ is assumed to remain invariant with respect to the reformulation. We have a finite set of *reformulation agents*

$$\mathcal{A} = \{a_1, \ldots, a_m\}.$$

Here, we should mention that all $p \in \mathcal{P}$ contain nonlinear patterns that can be targeted by $a \in \mathcal{A}$. For each agent $a \in \mathcal{A}$ the pattern set $\Pi_a$ describes the patterns of nonlinearities that $a$ can reformulate.

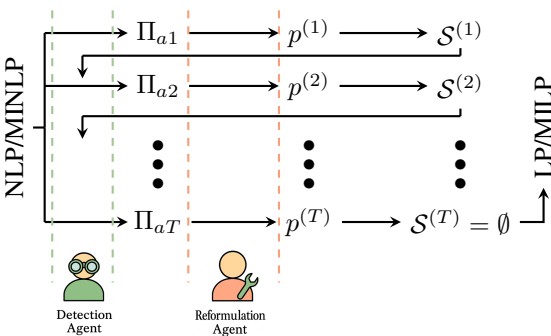

Figure 1: Illustration of the LinearizeLLM workflow

The function

$$\Lambda : \mathcal{P} \times \Theta \longrightarrow 2^{\mathcal{A}}, \qquad (p, \theta_p) \longmapsto \Lambda(p, \theta_p)$$

returns the subset of agents applicable to the pair $(p, \theta_p)$. At iteration $t$ $(t = 0, 1, \ldots, T)$ we abbreviate

$$\mathcal{S}^{(t)} := \Lambda\big(p^{(t)}, \theta_p\big) \subseteq \mathcal{A}.$$

The loop terminates at $T$ when $\mathcal{S}^{(T)} = \emptyset$, yielding the fully reformulated problem $p^{(T)}$.

For each $a \in \mathcal{A}$ we define the detection and reformulation operators

$$\mathrm{Detect}_a : \mathcal{P} \times \Theta \to 2^{\Pi_a}, \qquad \Phi_a : \mathcal{P} \times \Theta \times 2^{\Pi_a} \to \mathcal{P},$$

where $\mathrm{Detect}_a$ returns the currently active nonlinear pattern instances of type $a$, and $\Phi_a$ replaces a chosen set of instances, producing a new problem $p' \in \mathcal{P}$. For clarity, let

$$\Pi_a^{(t)} := \mathrm{Detect}_a\big(p^{(t)}, \theta_p\big)$$

denote the set of active instances of type $a$ at iteration $t$.

---

**Algorithm 1** *LinearizeLLM* pseudocode workflow

---

**Require:** Initial problem $p^{(0)}$, parameter set $\Theta$ with $\theta_p \in \Theta$
1: $\mathcal{S}^{(0)} \leftarrow \Lambda\big(p^{(0)}, \theta_p\big)$
2: $t \leftarrow 0$
3: **while** $\mathcal{S}^{(t)} \neq \emptyset$ **do**
4:     Choose $a_t \in \mathcal{S}^{(t)}$
5:     $\Pi_{a_t}^{(t)} \leftarrow \mathrm{Detect}_{a_t}\big(p^{(t)}, \theta_p\big)$
6:     $p^{(t+1)} \leftarrow \Phi_{a_t}\big(p^{(t)}, \theta_p, \Pi_{a_t}^{(t)}\big)$
7:     $\mathcal{S}^{(t+1)} \leftarrow \Lambda\big(p^{(t+1)}, \theta_p\big)$
8:     $t \leftarrow t + 1$
9: **end while**
10: **return** $p^{(T)}$ {fully reformulated LP/MILP}

---

The main components of the *LinearizeLLM* workflow are illustrated in Figure 1, and Algorithm 1 contains the pseudocode. First, $\Lambda$ is evaluated for the original nonlinear optimization problem $p^{(0)}$ together with its parameter set $\theta_p$. The result is the initial set of *reformulation agents* $\mathcal{S}^{(0)} \subseteq \mathcal{A}$ that are applicable to $p^{(0)}$. If $\mathcal{S}^{(0)} = \emptyset$ before entering the loop, then $p^{(0)}$ is either already linear (LP/MILP) or contains nonlinearities outside the scope of the agent set $\mathcal{A}$. A loop counter $t$ is initialized to zero. While at least one agent remains applicable, i.e., while $\mathcal{S}^{(t)} \neq \emptyset$, the algorithm chooses a single agent $a_t$ from $\mathcal{S}^{(t)}$. Once an agent has been selected, the detection routine $\mathrm{Detect}_{a_t}$ scans $p^{(t)}$ and identifies all yet-unreformulated nonlinear pattern instances of the type handled by $a_t$.

The reformulation operator $\Phi_{a_t}$ then replaces every instance in $\Pi_{a_t}^{(t)}$ with its exact linear counterpart and, where necessary, introduces auxiliary variables and constraints. The outcome of this transformation is the updated optimization problem $p^{(t+1)}$, in which the specific nonlinearity pattern targeted by $a_t$ no longer occurs. Because the structure of the optimization problem has changed, the set of applicable agents must be recomputed. A fresh call to $\Lambda$ using $p^{(t+1)}$ and $\theta_p$ yields the next pool $\mathcal{S}^{(t+1)}$. By construction, the agent $a_t$ just applied can no longer detect any instance of the pattern it targets and is therefore absent from $\mathcal{S}^{(t+1)}$. The loop counter is then incremented, and the procedure repeats until no agent remains applicable. Each pass through the loop eliminates at least one active nonlinearity instance, and no agent reintroduces a previously handled instance; hence the total number of unresolved nonlinearities strictly decreases. Consequently, the loop terminates after a finite number $T$ of iterations with $\mathcal{S}^{(T)} = \emptyset$, at which point the algorithm returns the fully linear (MILP or LP) model $p^{(T)}$.

# 6 EXPERIMENTS

## 6.1 DATASETS

The problems were derived from the ComplexOR dataset (Xiao et al., 2024). We produced 20 problem files by injecting nonlinearities into selected dataset instances. The problem descriptions, mathematical models, parameters, and Gurobi files were updated accordingly. Every revised instance comes with a Gurobi model that leverages the solver's built-in reformulation for the relevant nonlinear constraints and can therefore serve as a verification artifact. Table 1 summarizes the nonlinearity patterns that arise across the 20 files. Note that 2 instances contain 2 nonlinearity patterns. The detailed problem instances can be found in Appendix A.1.

Table 1: Nonlinearity patterns present in the 20 modified COMPLEXOR problem files.

| Nonlinearity | Count | Nonlinearity | Count | Nonlinearity | Count |
|---|---|---|---|---|---|
| Bilinear terms | 6 | Min operator | 3 | Max operator | 4 |
| Absolute-value | 4 | Linear fractional | 3 | Monotone transformations | 2 |

## 6.2 EXPERIMENTAL SETUP

Our testing framework is based on a multi-stage pipeline that converts mathematical optimization problems from LATEX format into executable Python code. The framework reads in the raw LATEX file containing the nonlinear optimization problem along with associated parameter and decision variable lists. The system implements a multi-stage processing pipeline. **Problem extraction:** the optimization problem written in LATEX together with its associated parameters is loaded. **Reformulation and detection pipeline:** we then invoke the core loop of our framework as defined in Algorithm 1, where a *detection agent* identifies nonlinear patterns and a *reformulation agent* $a_t \in \mathcal{A}$ is randomly selected to linearize them, with random selection helping to avoid bias toward any single reformulation path.

Specifically, we target $m = 6$ nonlinear patterns, each of which admits an exact reformulation into an LP or MILP:

**Bilinear interactions**: any product of two (possibly distinct) decision–variable expressions, $v_1(\mathbf{x})\, v_2(\mathbf{x})$, where each $v_\ell(\mathbf{x})$, $\ell = 1, 2$, may be binary and at most one may be continuous. If continuous variables appear in a bilinear interaction, we assume these variables are bounded in order to derive exact reformulations. For the remainder of this manuscript, references to bilinear interactions exclusively denote binary–binary and binary–continuous products of decision variables.

**Minimum operator**: the pointwise minimum of a finite family of linear functionals, $\min_{c \in \mathcal{C}}\{ f_c(\mathbf{x}) \}$, $\quad \mathcal{C} = \{1, \ldots, C\}$.

**Maximum operator**: the pointwise maximum of a finite family of linear functionals, $\max_{c \in \mathcal{C}}\{ f_c(\mathbf{x}) \}$, $\quad \mathcal{C} = \{1, \ldots, C\}$.

**Absolute-value operator**: the absolute-value of a linear functional, $|f(\mathbf{x})|$.

**Linear fractional**: a ratio of two affine expressions, $\frac{a^\top \mathbf{x}+b}{c^\top \mathbf{x}+d}$, $\quad c^\top \mathbf{x}+d > 0$ on the feasible region.

**Monotone transformations**: application of a strictly monotone function $\varphi \colon \mathbb{R} \to \mathbb{R}$, $\varphi\big(g(\mathbf{x})\big)$, which preserves the ordering of objective values or feasibility when $\varphi$ is applied coherently to both sides of a constraint. In order to recover the original objective value, the agent is instructed to apply the inverse transformation $\varphi^{-1}$.

All prompts related to detection and reformulation of nonlinear patterns and experimental setup can be found in Appendix A.2. Furthermore, detailed description of the exact linearization techniques applied by the *reformulation agents*, we refer the reader to Appendix A.3. The output of Algorithm 1 is the fully reformulated problem $p^{(T)}$ which is then translated into Python code by the LLM, following the principles of the Chain-of-Experts framework proposed by (Xiao et al., 2024). The resulting code is executed using Gurobi's Python interface. We conduct each experiment using a sampling `temperature` $\in [0, 0.15]$ and nucleus sampling `top-p` $\in [0.9, 1.0]$.

### 6.3 PERFORMANCE METRICS

Since manually verifying the LLM-generated reformulations for each optimization problem is infeasible at scale, we introduce four quantitative metrics to evaluate the performance of the *LinearizeLLM* framework:

**Detection Success Rate (DSR):** This metric measures the proportion of instances in which the correct nonlinearity pattern is successfully identified by the *detection agents*. Each problem instance is annotated with the specific nonlinearity pattern that *LinearizeLLM* is intended to reformulate.

**Reformulation Success Rate (RSR):** This metric assesses whether the *reformulation agents* were able to successfully produce a valid LP/MILP formulation of the original nonlinear problem. A reformulation is considered unsuccessful (i.e., a *reformulation error*) if either the final model is not classified as an LP or MILP by Gurobi or the model is infeasible or unbounded when solved.

**Compiler Success Rate (CSR):** This metric captures the success rate of compiling the LaTeX-formulated optimization problems into executable code. Although less central to our evaluation since we focus on specific nonlinearity patterns it is included for completeness. Specifically, when reformulation success cannot be conclusively verified.

**Overall Success Rate (OSR):** Our overall performance measure, OSR denotes the proportion of total runs that are free of detection, reformulation, or compilation errors and yield a reformulated model whose optimal objective value matches (with a tolerance of $\epsilon = 10^{-4}$) that of the original nonlinear problem, $p^{(0)}$, as solved by Gurobi. A tolerance of $10^{-4}$ accounts for numerical precision differences introduced during reformulation, such as auxiliary variables, while still validating that the reformulated problems preserve the essential structure and solution quality of the original formulations.

Each experiment is evaluated over five independent runs to ensure robustness, and results are averaged across these trials.

## 7 RESULTS

We begin our empirical analysis with a head-to-head baseline comparison, first evaluating the performance of *LinearizeLLM (Gemini 2.5 Flash)* against *LinearizeLLM (o3)* and then contrasting it with a one-shot prompt using *Gemini 2.5 Flash* itself under the full-context setting. This allows us to identify the strongest approach, which we subsequently use as the reference point for further analysis. In the next step, we dissect its behavior under different levels of problem context, showing that additional information does not uniformly improve performance. Further details on the reproducibility of results can be found in Appendix A.4.

## 7.1 HEAD-TO-HEAD BASELINE COMPARISON

We introduce a *one-shot* baseline that places the entire task (detection → reformulation) into a single prompt. The one-shot prompt can be found in Appendix A.2. To ensure fairness, we use the same decoding and context parameters as for *LinearizeLLM (Gemini 2.5 Flash)*.

| | vs. *LinearizeLLM (o3)* | | | | | vs. One-shot (*Gemini 2.5 Flash*) | | | |
|---|---|---|---|---|---|---|---|---|---|
| | $\triangle$**OSR** | $\triangle$**DSR** | $\triangle$**RSR** | $\triangle$**CSR** | | $\triangle$**OSR** | $\triangle$**DSR** | $\triangle$**RSR** | $\triangle$**CSR** |
| Bilin. | +58.7 | +0.0 | +20.5 | +20.5 | Bilin. | +3.1 | +0.0 | +0.0 | +3.1 |
| Min | +49.3 | +0.0 | +49.3 | +0.0 | Min | +49.3 | +0.0 | +0.0 | +0.0 |
| Max | +275.0 | +33.3 | +50.0 | −10.5 | Max | +15.4 | +11.1 | +5.9 | −15.0 |
| Abs. val. | +33.3 | +0.0 | +33.3 | +0.0 | Abs. val. | +33.3 | +0.0 | +0.0 | +0.0 |
| Lin. frac. | +75.5 | +25.0 | +0.0 | +14.9 | Lin. frac. | +6.9 | +0.0 | +14.9 | +0.0 |
| Mon. transf. | +150.0 | +0.0 | +42.9 | +42.9 | Mon. transf. | +25.0 | +0.0 | +0.0 | +0.0 |
| **Mean** | **+107.0** | **+9.7** | **+32.7** | **+11.3** | **Mean** | **+22.2** | **+1.9** | **+3.5** | **−2.0** |

Table 2: Relative gain/loss of *LinearizeLLM (Gemini 2.5 Flash)* over two baselines. Cells report $\Delta\% = 100 \times (\text{metric}_{\textbf{LinLLMGemini}} - \text{metric}_{\text{baseline}})/\text{score}_{\text{baseline}}$. Positive values mean *LinearizeLLM (Gemini 2.5 Flash)* outperforms the baseline; negative values mean the baseline is better.

***LinearizeLLM (Gemini 2.5 Flash)* vs. *LinearizeLLM (o3)*.** *LinearizeLLM (Gemini 2.5 Flash)* outperforms *LinearizeLLM (o3)* on nearly every metric and pattern (Table 2, left). We attribute this to three factors that align with prior evidence on multi-step prompting and tool-use reliability: (i) *pattern detection robustness*: *Gemini 2.5 Flash* exhibits fewer misclassifications, (ii) *bound/typing fidelity*: fewer failures in variable/parameter typing and bound derivation, which directly affect bilinear tightness and feasibility checks, and (iii) *code-generation stability*: fewer compile-time issues when emitting solver code.

***LinearizeLLM (Gemini)* vs. One-shot *(Gemini)*.** The one-shot baseline underperforms *LinearizeLLM* because since a single pass increases compiler and consistency errors (Table 2, right). For the **min** operator, instances with two nonlinearities (**min + absolute value**) led one-shot baseline to produce no tight reformulations $(0/5)$, sharply reducing OSR. For the **max** operator, about two thirds of our cases place the max-term directly in the objective. In such cases one must use an *epigraph* reformulation: for an objective $\min_x \max_k g_k(x)$ introduce an auxiliary $z$ and solve $\min_{x,z} z$ subject to $z \geq g_k(x)$ for all $k$. This two-step encoding (new variable + linking constraints) seem error-prone in a single pass; both approaches struggle on two specific instances with objective max-terms, but *LinearizeLLM* resolves more of them via staged detection and reformulation application, yielding higher OSR overall. For **absolute value**, the difficult cases are those with *multiple* nonlinearities in the same instance (e.g., absolute-value term combined with a second nonlinearity). Here the one-shot prompt tends to under-specify the necessary modeling steps, which lowers OSR; the stepwise checks in *LinearizeLLM* better manage this interaction complexity. For **linear fractional** terms, consider the starting form $\min_x(a^\top x + b)/(c^\top x + d)$ (or an equivalent constraint), with $c^\top x + d > 0$. One-shot often performs only a partial transformation: it introduces an objective variable $u$ for the fraction and then *cross-multiplies* $u(c^\top x + d) \geq a^\top x + b$, leaving the bilinear term $u\,x$ so the model remains nonlinear. The exact Charnes–Cooper transform avoids this by setting $t = 1/(c^\top x + d) > 0$, $y = xt$, enforcing $c^\top y + dt = 1$, and rewriting $u \geq a^\top y + bt$, which is fully linear. For **monotone transformations**, OSR drops when one-shot omits the inverse mapping after solving (e.g., minimize $\exp(s)$ by minimizing $s$, but then neglect to report back $\exp(s^\star)$). Finally, for **bilinear** terms, the gap is smaller but still favors *LinearizeLLM* thanks to explicit bound derivation that avoids loose or missing $M$ values. Overall, the staged workflow of *LinearizeLLM* delivers higher OSR and RSR, while one-shot prompting is limited by compilation errors and incomplete transformations.

Having established that *Gemini 2.5 Flash* within the *LinearizeLLM* framework provides the most reliable baseline, we next turn to a context-blind ablation study to investigate how varying levels of problem context (**full**, **partial**, or **no context**) influence its detection, reformulation and overall performance.

## 7.2 CONTEXT-BLIND ABLATION STUDY

The Context-Blind Ablation study examines how varying levels of problem context affect the LLM agents' ability to detect nonlinear patterns and perform LP/MILP reformulations. We perform our experiments with the *Gemini 2.5 Flash* model. We evaluate *LinearizeLLM* under three distinct information scenarios: (i) **No Context**: Only the raw LaTeX mathematical formulation is provided. No decision variable or parameter definitions are given. (ii) **Partial Information**: The LaTeX formulation plus decision variable definitions are provided. However, no concrete parameter definitions are available. (iii) **Full Information**: Complete problem context is given.

| | FULL | | | | PARTIAL | | | | NO CONTEXT | | | |
|---|---|---|---|---|---|---|---|---|---|---|---|---|
| **Bilin.** | 100% | 100% | 100% | 100% | 100% | 100% | 100% | 100% | 93% | 97% | 100% | 100% |
| **Min** | 100% | 100% | 100% | 100% | 100% | 100% | 100% | 100% | 87% | 100% | 93% | 93% |
| **Max** | 75% | 100% | 90% | 85% | 75% | 100% | 100% | 100% | 80% | 100% | 100% | 100% |
| **Absolute val.** | 100% | 100% | 100% | 100% | 100% | 100% | 100% | 100% | 85% | 100% | 95% | 90% |
| **Lin. frac.** | 93% | 100% | 100% | 100% | 87% | 100% | 100% | 87% | 100% | 100% | 100% | 100% |
| **Mon. transf.** | 100% | 100% | 100% | 100% | 100% | 100% | 100% | 100% | 100% | 100% | 100% | 100% |
| | OSR | DSR | RSR | CSR | OSR | DSR | RSR | CSR | OSR | DSR | RSR | CSR |

Figure 2: Heat-map of OSR, DSR, RSR and CSR for six nonlinearity patterns when *Gemini 2.5 Flash* is prompted with **full**, **partial** or **no context** (left to right blocks). Darker cells indicate higher percentages.

Interestingly, additional context does not always improve performance. In Figure 2 for **bilinear** terms, missing type information leads the detection agent to misclassify parameter–variable products as bilinear. In addition, no tight bounds are applied to reformulate continuous–binary products exactly, which result in non-tight reformulations and thus a drop in OSR. For the **min** operator, the drop in OSR stems from failures to reformulate the pattern into a valid LP/MILP, with additional compilation errors during code execution further amplifying the decline of OSR. For the **max** operator, in about two-thirds of cases the term appears directly in the objective, requiring an auxiliary variable and epigraph reformulation. In these instances, richer descriptions of decision variables and parameters can trigger over-aggregation or parsing noise, an effect consistent with prior observations that LLMs may become sensitive to prompt format and degrade when exposed to redundant context (Wei et al., 2022). Simpler forms may often yield cleaner detection—explaining the occasional gains observed under no context. For the **absolute value** pattern under the no-context setting, we observe a marked drop in OSR driven by declines in both RSR and CSR. This can be traced to cases where two nonlinearities, a **min** and an **absolute value**, appear jointly, making correct reformulation and compilation substantially more difficult. **Linear fractional** losses stem from cases where added context causes fallback to a non-tight reformulation instead of exact Charnes–Cooper transformation into a LP/MILP. By contrast, **monotone transformations** remain stable since once monotonicity is identified, reformulation is straightforward.

## 8 CONCLUSION

*LinearizeLLM* turns exact nonlinear-to-linear reformulation into an automated, auditable step. Pattern-specific *reformulation agents* detect each nonlinearity pattern, apply textbook transformations, and hand the result to any LP/MILP solver through a simple detect–reformulate loop. On a benchmark dataset of 20 nonlinear problems derived from COMPLEXOR, OpenAI's *o3* and Google's *Gemini 2.5 Flash* reliably produced equivalent linear models. *LinearizeLLM* with *Gemini 2.5 Flash* secures perfect detection and reformulation success on four of the six nonlinear-patterns, and even on the harder nonlinearity patterns still reaches more than 75% while boosting overall success rate by up to 107% compared with the benchmarks. Collectively, these figures show that the model handles virtually every benchmark instance across diverse nonlinearities. Overall, modest agentic decomposition plus targeted prompting lets general-purpose LLMs deliver solver-ready linear models, bringing fully conversational nonlinear optimization within reach.

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

# A  APPENDIX

## A.1  PROBLEM INSTANCES

### A.1.1  AIRCRAFT PROBLEM

The Aircraft Landing Problem (ALP) is the problem of deciding a landing time on an appropriate runway for each aircraft in a given set of aircraft such that each aircraft lands within a predetermined time window; and separation criteria between the landing of an aircraft, and the landing of all successive aircraft, are respected. We are given the earliest landing time, latest landing time, target landing time, and penalties for landing before or after the target landing time for each aircraft. There is also a separation time that represents the minimum time required between the landing of two successive aircraft.

**Compact formulation with explicit index ranges.**

$$
\min_{t,e,l} \ \sqrt{\left( \sum_{i=1}^{n} (\alpha_i\, e_i + \beta_i\, l_i) \right)}
$$

$$
\begin{aligned}
\text{s.t. } & E_i \ \le\ t_i \ \le\ L_i && \forall\, i = 1,\ldots,n \\
& t_i - T_i \ =\ l_i - e_i && \forall\, i = 1,\ldots,n \\
& t_i \ \le\ t_{i+1} && \forall\, i = 1,\ldots,n-1 \\
& t_{i+1} - t_i \ \ge\ \max\!\big(S_{i,i+1},\, l_i\big) && \forall\, i = 1,\ldots,n-1 \\
& e_i,\, l_i \ \ge\ 0 && \forall\, i = 1,\ldots,n
\end{aligned}
$$

### A.1.2  BLEND PROBLEM #1

The problem aims to determine the optimal amounts of alloys to purchase in order to achieve a desired blend of required elements at the minimum cost. We are given a set of alloys available on the market and a set of required elements for the blend, the percentage composition data of each required element in each alloy, the desired blend percentage of each required element, the price of each alloy. The decision is the amount of each alloy to be purchased, which is continuous. The objective is to minimize the total cost of the alloy purchased. There are two constraints. The first set of constraints ensures that the desired blend percentage of each required element is met. The second constraint ensures that the total amount of alloys purchased is equal to 1. Besides minimising purchase cost we penalise any departure from the desired composition. A weighted absolute deviation between the achieved and target percentage of each element is added to the objective.

**Compact formulation with explicit index ranges.**

$$\min_x \; \sum_{j=1}^{J} p_j x_j \; + \; \sum_{i=1}^{I} \lambda_i \left| \sum_{j=1}^{J} c_{ij} x_j - d_i \right|$$

$$\text{s.t.} \; \sum_{j=1}^{J} x_j = 1$$

$$x_j \geq 0 \qquad\qquad\qquad \forall j = 1, .., J$$

### A.1.3 BLEND PROBLEM #2

The problem asks for the best way to blend three raw materials to produce exactly one tonne of flame-retardant resin so as to minimize the average cost per kilocalorie of heat generated. Each material's characteristics are known: its cost in dollars per kilogram, its heat content in kilocalories per kilogram and its chlorine concentration by weight. The final resin must provide at least 7 200 kcal in total.

**Compact formulation with explicit index ranges.**

$$\min_x \; \frac{\sum_{i=1}^{3} c_i x_i}{\sum_{i=1}^{3} h_i x_i}$$

$$\text{s.t.} \; \sum_{i=1}^{3} h_i x_i \; \geq \; 7200$$

$$\sum_{i=1}^{3} x_i \; = \; 1000$$

$$x_i \; \geq \; 0 \qquad (i = 1, 2, 3).$$

### A.1.4 DIET PROBLEM #1

Consider a diet problem. Given a set of foods $\mathcal{F} = \{1, \ldots, J\}$ and nutrients $\mathcal{N} = \{1, \ldots, I\}$, let $p_j$ be the unit price of food $j$ and $x_j$ the quantity to buy. For each nutrient $i$, $a_{ij}$ is the amount of nutrient $i$ in one unit of food $j$ and $m_i$ and $M_i$ are the lower and upper recommended intakes. The nutritionist wants the **total cost** $\sum_{j=1}^{J} p_j x_j$ to be as close as possible to a target budget $B$, while every nutrient intake stays inside its recommended band and every food quantity stays in its allowable range.

**Compact formulation with explicit index ranges.**

$$\min_x \; \left| \sum_{j=1}^{J} p_j x_j \; - \; B \right|$$

$$\text{s.t.} \; \sum_{j=1}^{J} a_{ij} x_j \; \geq \; \min\!\left( m_i, \; \sum_{j=1}^{J} a_{1j} x_j \right) \quad \forall i = 1, \ldots, I$$

$$\sum_{j=1}^{J} a_{ij} x_j \; \leq \; M_i \qquad\qquad\qquad \forall i = 1, \ldots, I$$

$$\underline{x}_j \; \leq \; x_j \; \leq \; \overline{x}_j \qquad\qquad\qquad \forall j = 1, \ldots, J$$

### A.1.5   DIET PROBLEM #2

The *Diet Problem* asks us to decide how much of each food $j \in \{1, \dots, J\}$ to buy so that all required nutrients are consumed within specified ranges while exponentially scaled costs are minimized. Each food $j$ has

- a unit cost $p_j$,
- lower and upper purchase bounds $\left[ \underline{x}_j, \, \overline{x}_j \right]$.

Each nutrient $i \in \{1, \dots, I\}$ must be obtained in an amount lying inside the interval $\left[ m_i, \, M_i \right]$. The quantity of nutrient $i$ contained in one unit of food $j$ is $a_{i,j}$.

**Compact formulation with explicit index ranges.**

$$\min_x \; exp\left( \sum_{j=1}^{J} p_j\, x_j \right)$$

$$\text{s.t. } \underline{x}_j \; \leq \; x_j \; \leq \; \overline{x}_j \qquad\qquad \forall\, j = 1, \dots, J$$

$$m_i \; \leq \; \sum_{j=1}^{J} a_{i,j}\, x_j \; \leq \; M_i \quad \forall\, i = 1, \dots, I$$

$$x_j \; \geq \; 0 \qquad\qquad\qquad \forall\, j = 1, \dots, J$$

### A.1.6   DIET PROBLEM #3

Consider a diet problem. Given a set of nutrients *Nutrients* and a set of foods *Foods*. Each food $j$ has a cost $Cost_j$ and a range of amount that can be bought $[MinAmount_j, \, MaxAmount_j]$. Each nutrient $i$ has a range of amount that should be included in the diet $[MinNutrient_i, \, MaxNutrient_i]$. The amount of nutrient $i$ in food $j$ is $NutrientAmount_{i,j}$.

The problem aims to minimize the average cost per gram of food, i.e. the ratio of total cost to total quantity purchased. It is constrained that the total intake of each nutrient $i$ must lie within its recommended range $[MinNutrient_i, \, MaxNutrient_i]$. Moreover, because micronutrient uptake is enhanced by vitamin-C intake, each nutrient must be consumed in an amount at least the smaller of its own nominal minimum requirement and the diet's total vitamin-C content.

**Compact formulation with explicit index ranges.**

$$\min_x \; \frac{\displaystyle\sum_{j=1}^{J} p_j\, x_j}{\displaystyle\sum_{j=1}^{J} x_j}$$

$$\text{s.t. } \sum_{j=1}^{J} a_{ij}\, x_j \; \leq \; M_i \quad \forall\, i = 1, \dots, I$$

$$\underline{x}_j \; \leq \; x_j \; \leq \; \overline{x}_j \quad \forall\, j = 1, \dots, J$$

$$\sum_{j=1}^{J} x_j \; \geq \; Q_{\min} \quad \text{(minimum total quantity)}$$

### A.1.7   KNAPSACK PROBLEM #1

The Knapsack Problem is a classic optimization problem in operations research and computer science. The problem is to determine the most valuable combination of items to include in a knapsack,

given a set of items with different values and weights, and a maximum weight capacity of the knapsack. The goal is to maximize the total value of the items in the knapsack without exceeding its weight capacity. Interpreting "weight" as a volume constraint, we account for nesting one item inside another, say, socks in shoes, by subtracting a fixed 10-unit volume reduction whenever both items are selected.

**Compact formulation with explicit index ranges.**

$$\max_{x} \quad \sum_{j=1}^{J} v_j \, x_j$$

$$\text{s.t.} \quad \sum_{j=1}^{J} w_j \, x_j - 10 \min\{x_1, \, x_3\} \; \leq \; W,$$

$$x_j \; \in \; \{0, 1\} \quad \forall j = 1, \ldots, J.$$

### A.1.8 KNAPSACK PROBLEM 2

The Knapsack Problem is a classic optimization problem in operations research and computer science. The problem is to determine the most valuable combination of items to include in a knapsack, given a set of items with different values and weights, and a maximum weight capacity of the knapsack. The goal is to maximize the total value of the items in the knapsack without exceeding its weight capacity. Due to synergy, an additional value of $b \min(x, y)$ is obtained when items $x$ and $y$ are used together.

**Compact formulation with explicit index ranges.**

$$\max_{x} \quad \sum_{j=1}^{J} v_j \, x_j + b \min\{x_1, \, x_2\}$$

$$\text{s.t.} \quad \sum_{j=1}^{J} w_j \, x_j \; \leq \; W,$$

$$x_j \; \in \; \{0, 1\} \quad \forall j = 1, \ldots, J.$$

### A.1.9 MEDIA SELECTION PROBLEM #1

The main media selection problem is a problem of allocating advertising budgets between possible advertising outlets. Given a set of media options, it aims to determine which media should be selected so that all audiences are reached with minimum campaign cost. It does not matter if an audience is covered more than once, as long as it is covered at least once. Moreover, the company does not wish to spend more money on the campaign than necessary. Due to synergy, a discount $d$ is applied when two certain medias are used simultaneously.

**Compact formulation with explicit index ranges.**

$$\min_{x} \quad \sum_{m=1}^{3} c_m \, x_m - d * x_1 * x_2$$

$$\text{s.t.} \quad \sum_{m=1}^{3} a_{tm} \, x_m \; \geq \; m_t \qquad \forall t = 1, \ldots, 3,$$

$$x_m \; \in \; \{0, 1\} \qquad \forall m = 1, \ldots, 3.$$

A.1.10 MEDIA SELECTION PROBLEM #2

The main media selection problem is a problem of allocating advertising budgets between possible advertising outlets. Given a set of media options, it aims to determine which media should be selected so that all audiences are reached with minimum campaign cost. It does not matter if an audience is covered more than once, as long as it is covered at least once. Moreover, the company does not wish to spend more money on the campaign than necessary. Additional, limited resources can be used for a quality increase of the campaign, denoted by $q_m$. This decreases cost by $r_m$ per money spent on this quality increase.

**Compact formulation with explicit index ranges.**

$$\min_{x,q} \quad \sum_{m=1}^{3} \left( c_m x_m - r_m x_m q_m \right)$$

$$\text{s.t.} \quad \sum_{m=1}^{3} a_{tm} x_m \geq m_t \qquad \forall t = 1, \dots, 3,$$

$$\sum_{m=1}^{3} q_m \leq 5$$

$$x_m \in \{0,1\} \qquad \forall m = 1, \dots, 3$$

$$q_m \geq 0 \qquad \forall m = 1, \dots, 3$$

A.1.11 MULTI PROBLEM #1

This is a multi-commodity transportation problem. Given a set of origins Origins, a set of destinations Destinations, and a set of products Products. Each origin $i$ has a certain supply of each product $p$, denoted $\text{Supply}_{i,p}$, and each destination $j$ has a certain demand for each product $p$, denoted $\text{Demand}_{j,p}$. The cost of shipping one unit of product $p$ from origin $i$ to destination $j$ is $c_{i,j,p}$. The shipment is allowed to deviate from the demand. However, this incurs cost, proportional to the deviation times a constant $r$. The problem aims to minimize the total cost of shipping all products from the origins to the destinations plus the costs from deviations. It is constrained that

- the total amount of each product $p$ shipped from each origin $i$ equals its supply $\text{Supply}_{i,p}$,
- the total amount of all products shipped from each origin $i$ to each destination $j$ does not exceed a certain limit $\text{Limit}_{i,j}$.

How to decide the number of units $x_{i,j,p}$ of each product $p$ to be shipped from each origin $i$ to each destination $j$?

**Compact formulation with explicit index ranges.**

$$\min_{x} \quad \sum_{i=1}^{2}\sum_{j=1}^{2}\sum_{p=1}^{2} c_{i,j,p}\, x_{i,j,p} + \sum_{j=1}^{2}\sum_{p=1}^{2} r \left| \sum_{i=1}^{2} x_{i,j,p} - \text{Demand}_{j,p} \right|$$

$$\text{s.t.} \quad \sum_{j=1}^{2} x_{i,j,p} = \text{Supply}_{i,p} \qquad\qquad \forall i = 1, 2,\ \forall p = 1, 2,$$

$$\sum_{p=1}^{2} x_{i,j,p} \leq \text{Limit}_{i,j} \qquad\qquad \forall i = 1, 2,\ \forall j = 1, 2,$$

$$x_{i,j,p} \geq 0 \qquad\qquad \forall i, j, p.$$

A.1.12 NETASGN PROBLEM #1

Consider a project assignment problem. Given a set of people *People* and a set of projects *Projects*. Each person $i$ has a certain number of available hours $S_i$ and each project $j$ requires a certain number

of hours $D_j$. The cost per hour of work for person $i$ on project $j$ is $c_{ij}$. Each person $i$ can contribute to project $j$ up to a maximum limit $\ell_{ij}$. The problem aims to minimise the total cost of assigning people to projects. To ensure fairness, we also include a penalty proportional to the maximum pairwise deviation in total hours worked—namely $r$ times that deviation. It is constrained that the total number of hours assigned from each person $i$ equals its supply $S_i$ and the total number of hours assigned to each project $j$ equals its demand $D_j$. How to decide the number of hours $x_{ij}$ to be assigned from each person $i$ to each project $j$?

**Compact formulation with explicit index ranges.**

$$\min_x \quad \sum_{i=1}^{2}\sum_{j=1}^{2} c_{ij}\,x_{ij} + r\left|\sum_{j=1}^{2} x_{1j} - \sum_{j=1}^{2} x_{2j}\right|$$

$$\text{s.t.} \quad \sum_{j=1}^{2} x_{ij} = S_i \qquad\qquad \forall\, i = 1, 2,$$

$$\sum_{i=1}^{2} x_{ij} = D_j \qquad\qquad \forall\, j = 1, 2,$$

$$0 \leq x_{ij} \leq \ell_{ij} \qquad\qquad \forall\, i = 1, 2,\ j = 1, 2.$$

### A.1.13 NETASGN PROBLEM #2

Consider a project assignment problem. Given a set of people *People* and a set of projects *Projects*. Each person $i$ has a certain number of available hours $S_i$ and each project $j$ requires a certain number of hours $D_j$. The cost per hour of work for person $i$ on project $j$ is $c_{ij}$. Each person $i$ can contribute to project $j$ up to a maximum limit $\ell_{ij}$. The problem aims to minimise the total cost of assigning people to projects. To ensure fairness, we also include a penalty proportional to the maximum pairwise deviation in total hours worked—namely $r$ times that deviation. It is constrained that the total number of hours assigned from each person $i$ equals its supply $S_i$ and the total number of hours assigned to each project $j$ equals its demand $D_j$. How to decide the number of hours $x_{ij}$ to be assigned from each person $i$ to each project $j$?

**Compact formulation with explicit index ranges.**

$$\min_x \quad \sum_{i=1}^{2}\sum_{j=1}^{2} c_{ij}\,x_{ij} + r\max\left\{\left(\sum_{j=1}^{2} x_{1j} - \sum_{j=1}^{2} x_{2j}\right),\ \left(\sum_{j=1}^{2} x_{2j} - \sum_{j=1}^{2} x_{1j}\right)\right\}$$

$$\text{s.t.} \quad \sum_{j=1}^{2} x_{ij} = S_i \qquad\qquad\qquad\qquad \forall\, i = 1, 2,$$

$$\sum_{i=1}^{2} x_{ij} = D_j \qquad\qquad\qquad\qquad \forall\, j = 1, 2,$$

$$0 \leq x_{ij} \leq \ell_{ij} \qquad\qquad\qquad\qquad \forall\, i = 1, 2,\ j = 1, 2.$$

### A.1.14 NETMCOL PROBLEM #1

Consider a transportation problem with multiple products. Given a set of cities $\mathcal{C}$ and a set of directed links $\mathcal{L} \subseteq \mathcal{C} \times \mathcal{C}$. Each city $i \in \mathcal{C}$ has a supply $Supply_{i,p}$ and a demand $Demand_{i,p}$ for each product $p \in \mathcal{P}$. Shipping one unit of product $p$ from city $i$ to city $j$ costs $ShipmentCost_{i,j,p}$. Each link $(i, j) \in \mathcal{L}$ has a per-product capacity $Capacity_{i,j,p}$ and a joint capacity $JointCapacity_{i,j}$ across all products. We choose shipment quantities $x_{i,j,p} \geq 0$ to satisfy supplies and demands at minimum total cost, respecting individual and joint link capacities. Through a special contract with a transportation company, variable $ShipmentCost_{i,j,p}$ costs can be reduced by 20%. The contract incurs fixed costs $ContractCosts_{i,j}$. The contract decision is modeled with a binary $z_{i,j}$.

**Compact formulation with explicit index ranges.**

$$\min_{x,z} \quad \sum_{(i,j)\in\mathcal{L}} \sum_{p\in\mathcal{P}} (1 - 0.2 z_{i,j})\, ShipmentCost_{i,j,p}\, x_{i,j,p} + z_{i,j} ContractCosts_{i,j}$$

$$\text{s.t.} \quad \sum_{j:\,(j,i)\in\mathcal{L}} x_{j,i,p} + Supply_{i,p} = \sum_{j:\,(i,j)\in\mathcal{L}} x_{i,j,p} + Demand_{i,p} \qquad \forall i \in \mathcal{C},\ p \in \mathcal{P},$$

$$x_{i,j,p} \leq Capacity_{i,j,p} \qquad \forall (i,j) \in \mathcal{L},\ p \in \mathcal{P},$$

$$\sum_{p\in\mathcal{P}} x_{i,j,p} \leq JointCapacity_{i,j} \qquad \forall (i,j) \in \mathcal{L},$$

$$x_{i,j,p} \geq 0 \qquad \forall (i,j) \in \mathcal{L},\ p \in \mathcal{P}$$

$$z_{i,j} \in \{0,1\} \qquad \forall (i,j) \in \mathcal{L}$$

### A.1.15 NETMCOL PROBLEM #2

Consider a transportation problem with multiple products. Given a set of cities $\mathcal{C}$ and a set of directed links $\mathcal{L} \subseteq \mathcal{C} \times \mathcal{C}$. Each city $i \in \mathcal{C}$ has a supply $Supply_{i,p}$ and a demand $Demand_{i,p}$ for each product $p \in \mathcal{P}$. Shipping one unit of product $p$ from city $i$ to city $j$ costs $ShipmentCost_{i,j,p}$. Each link $(i,j) \in \mathcal{L}$ has a per-product capacity $Capacity_{i,j,p}$ and a joint capacity $JointCapacity_{i,j}$ across all products. Demand cannot be exceeded, but need not be met. There is revenue $r_p$ associated with each unit of demand met. The operator chooses $x_{i,j,p} \geq 0$ and seeks to maximize revenue per shipment costs.

**Compact formulation with explicit index ranges.**

$$\max_{x,y} \quad \frac{\sum_{i\in\mathcal{C}} \sum_{p\in\mathcal{P}} r_p\, y_{i,p}}{\sum_{(i,j)\in\mathcal{L}} \sum_{p\in\mathcal{P}} ShipmentCost_{i,j,p}\, x_{i,j,p} + 0.01}$$

$$\text{s.t.} \quad \sum_{j:\,(j,i)\in\mathcal{L}} x_{j,i,p} + Supply_{i,p} = \sum_{j:\,(i,j)\in\mathcal{L}} x_{i,j,p} + y_{i,p} \quad \forall i \in \mathcal{C},\ p \in \mathcal{P},$$

$$0 \leq y_{i,p} \leq Demand_{i,p} \qquad \forall i \in \mathcal{C},\ p \in \mathcal{P},$$

$$x_{i,j,p} \leq Capacity_{i,j,p} \qquad \forall (i,j) \in \mathcal{L},\ p \in \mathcal{P},$$

$$\sum_{p\in\mathcal{P}} x_{i,j,p} \leq JointCapacity_{i,j} \qquad \forall (i,j) \in \mathcal{L},$$

$$x_{i,j,p} \geq 0 \qquad \forall (i,j) \in \mathcal{L},\ p \in \mathcal{P}.$$

### A.1.16 NTRANS PROBLEM #1

Consider a transportation problem. Given a set of origins $Origins$ and a set of destinations $Destinations$. Each origin $i$ has a certain supply of goods $S_i$, and each destination $j$ has a certain demand for goods $D_j$. The cost of shipping one unit of goods from origin $i$ to destination $j$ is $r_{ij}$. However, the number of units shipped cannot exceed the limit $\ell_{ij}$. The problem aims to minimize the total cost of shipping goods from the origins to the destinations. We decide on shipments $x_{ij}$ so as to satisfy supply and demand without violating arc-capacity limits, at minimum cost. A fixed investment, costing $C$ would reduce $r_{ij}$ to $\tilde{r}_{ij}$. The investment decision is modeled with binary $z$.

**Compact formulation with explicit index ranges.** Let $x_{ij}$ be the number of units shipped from origin $i$ to destination $j$. With $I = 2$ and $J = 2$, we have

$$\min_{x,z} \quad (1-z)\sum_{i=1}^{2}\sum_{j=1}^{2} r_{ij}\,x_{ij} + z\sum_{i=1}^{2}\sum_{j=1}^{2} \tilde{r}_{ij}\,x_{ij} + Cz$$

$$\text{s.t.} \quad \sum_{j=1}^{2} x_{ij} \;\leq\; S_i \qquad\qquad \forall\, i = 1, 2$$

$$\sum_{i=1}^{2} x_{ij} \;\geq\; D_j \qquad\qquad \forall\, j = 1, 2$$

$$0 \leq x_{ij} \;\leq\; \ell_{ij} \qquad\qquad \forall\, i, j = 1, 2$$

$$z \in \{0, 1\}$$

### A.1.17 NTRANS PROBLEM #2

Consider a transportation problem. Given a set of origins $Origins$ and a set of destinations $Destinations$. Each origin $i$ has a certain supply of goods $S_i$, and each destination $j$ has a certain demand for goods $D_j$. The cost of shipping one unit of goods from origin $i$ to destination $j$ is $r_{ij}$. Shipments beyond the limit $\ell_{ij}$ are allowed but incur a per-unit penalty of $c$. The problem aims to minimize the total cost of shipping goods from the origins to the destinations. We decide on shipments $x_{ij}$ so as to satisfy supply and demand without violating arc-capacity limits, at minimum cost.

**Compact formulation with explicit index ranges.** Let $x_{ij}$ be the number of units shipped from origin $i$ to destination $j$. With $I = 2$ and $J = 2$, we have

$$\min_{x_{ij}\geq 0} \quad \sum_{i=1}^{2}\sum_{j=1}^{2}\left(r_{ij}\,x_{ij} + c\max\{0,\, x_{ij} - \ell_{ij}\}\right)$$

$$\text{s.t.} \quad \sum_{j=1}^{2} x_{ij} \;\leq\; S_i \qquad\qquad \forall\, i = 1, 2$$

$$\sum_{i=1}^{2} x_{ij} \;\geq\; D_j \qquad\qquad \forall\, j = 1, 2$$

### A.1.18 PROD PROBLEM #1

Consider a problem where we have a set $P$. For each element $j \in P$, we have a parameter $a_j$, a parameter $c_j$, and a parameter $u_j$. We also have a global parameter $b$. We introduce a decision variable $x_j$ for each $j \in P$. The goal is to maximize the total profit, $\sum_{j\in P} c_j\,x_j$. The constraints are that the total weighted usage $\sum_{j\in P} \frac{1}{a_j}\,x_j$ cannot exceed $b$, and each $x_j$ must lie between $0$ and $u_j$. With a media campaign, incurring costs $C$, the profitability per $x_j$ can be increased to $\tilde{c}_j$. The investment decision is denoted with binary $z$.

**Compact formulation with explicit index ranges.**

$$\max_{x,z} \quad (1-z)\sum_{j=1}^{3} c_j\,x_j + z\sum_{j=1}^{3} \tilde{c}_j\,x_j - z*C$$

$$\text{s.t.} \quad \sum_{j=1}^{3} \frac{1}{a_j}\,x_j \ \le\ 4,$$

$$0 \ \le\ x_j \ \le\ u_j \quad \forall j = 1,2,3.$$

$$z \in \{0,\,1\}$$

### A.1.19   PROD PROBLEM #2

Consider a problem where we have a set $P$, parameters $a_j$, $c_j$ and $u_j$ for each $j \in P$, global parameters $b$ and $c$, decision variables $x_j$, and we maximize $\sum_{j\in P} c_j x_j$ subject to $0 \le x_j \le u_j \ \forall j \in P$, where each unit of $x_j$ consumes $\frac{1}{a_j}$ of a resource of capacity $b$ and any excess $\max\{0, \sum_{j\in P} \frac{1}{a_j} x_j - b\}$ is penalized at rate $c$.

**Compact formulation with explicit index ranges.**

$$\max_{x} \quad \sum_{j=1}^{3} c_j\,x_j - c\max\left\{0,\ \sum_{j=1}^{3} \frac{1}{a_j}*x_j - b\right\},$$

$$\text{s.t.} \quad 0 \ \le\ x_j \ \le\ u_j \quad \forall j = 1,2,3.$$

### A.1.20   REVENUE MAXIMIZATION PROBLEM

We have a set of flight legs (one-way non-stop flights) with a limited passenger capacity. According to market research, we defined a set of flight itineraries to sell as packages with a given price. For each package, we have an estimated demand. How many units of each package should we sell to maximize the revenue? We reserve the passenger seats according to the number of packages we want to sell. A marketing campaign with fixed costs $C$ increases revenue from $r_i$ to $\tilde{r}_i$. The investment decision is modeled with binary $z$.

**Compact formulation with explicit index ranges.**

$$\max_{x,z} \quad (1-z)\sum_{i=1}^{2} r_i\,x_i + z\sum_{i=1}^{2} \tilde{r}_i\,x_i - z*C$$

$$\text{s.t.} \quad \sum_{i=1}^{2} \delta_{ij}\,x_i \ \le\ c_j \qquad\qquad \forall j = 1,2,3,$$

$$\underline{x}_i \ \le\ x_i \ \le\ \overline{x}_i \qquad\qquad \forall i = 1,2,$$

$$x_i \in \mathbb{Z} \qquad\qquad \forall i = 1,2,$$

$$z \in \{0,\,1\}\,.$$

## A.2   PROMPT TEMPLATES

This section contains all the prompts used in the *LinearizeLLM* for pattern detection, reformulation, and code generation. The original markdown prompt files have been transformed into pure LaTeX code with proper mathematical formatting and structured documentation. The content is the same.

### A.2.1 PATTERN DETECTION PROMPT

**Role Definition**    You are an expert in identifying **nonlinear patterns** in mixed-integer optimization problems.

**Task Description**    Given a LaTeX optimization problem, detect and group similar indexed instances of these nonlinearities:

1. **Bilinear:** Products of decision variables (e.g., $x_i \cdot y_j$)

2. **Min:** Simple min operators (e.g., $\min(x_i, y_i)$)

3. **Max:** Simple max operators (e.g., $\max(x_i, y_i)$)

4. **Absolute Value:** Terms like $|x_i - y_i|$ with linear decision variables

5. **Quotient:** Linear fractional terms where DECISION VARIABLES appear in either numerator or denominator or both (e.g., $\frac{y_i}{x_i}$ where both $x_i$ and $y_i$ are decision variables)

6. **Monotone Transformation:** Objective functions of form $\min f(g(x))$ where $g(x)$ is linear and $f$ is monotone function (e.g., $\log(\sum_i x_i)$)

**Important Notes**

- If a category has no patterns, write "NONE" under that category

- Focus on PATTERNS that represent multiple similar terms, not individual instances

- Report grouped patterns succinctly

- Example grouping: $x_i \cdot y_j, \forall i, j \in A$

- If none found, explicitly state "NONE"

**Input Information**

- **LaTeX model:** {latex_model}

- **Parameter context:** {parameter_context}

- **Concrete parameters (IGNORE these when detecting nonlinearities):** {param_info}

**Output Format**

```
 NON-LINEARITIES DETECTED: [YES/NO]

BILINEAR_PATTERNS:
* [List each bilinear pattern]

MIN_PATTERNS:
* [List each min pattern]

MAX_PATTERNS:
* [List each max pattern]

ABSOLUTE_PATTERNS:
* [List each absolute value pattern]

QUOTIENT_PATTERNS:
* [List each quotient pattern]

MONOTONE_TRANSFORMATION_PATTERNS:
* [List each monotone transformation pattern in objective function]
```

**Important Guidelines**

- Provide ONLY mathematical formulations

- Do NOT use markdown code fences or additional explanations

### A.2.2 BILINEAR PATTERN PROMPT

**Role Definition**  You are an expert in reformulating **bilinear terms** in mixed-integer optimization problems.

**Task Description**  For each **bilinear term** appearing in the LaTeX optimization problem, complete steps A–E:

1. **IDENTIFY** – Quote exact bilinear terms and its index set

2. **EVALUATE** – Check methods (**stop at first Applicable=YES, Exact=YES**):

   (a) McCormick envelopes (convex hull, 4 inequalities)

   (b) Disjunctive (binary linking constraints)

   (c) Binary Big-M (**only if 1–2 fail**, derive tightest M explicitly)

   Summarize evaluation briefly (e.g., McCormick: Applicable=..., Exact=...)

3. **DERIVE M** – Clearly state M if used (else "n/a")

4. **FORMULATE** – Provide constraints in LaTeX; no other changes

5. **VERIFY** – Brief one-line justification

**Input Information**

- **LaTeX model:** {latex_model}

- **Pattern description (human hint):** {bilinear_pattern}

- **Concrete parameters (bounds, indices):** {param_info}

**Output Format**

```
REPORT:
Pattern      : <bilinear expression and indices>
Technique    : <chosen method>
Verification : <one concise sentence>
Bounds / M   : <bounds or Big-M; "n/a" if not used>
Aux vars     : <any new variables; minimal>
```

**Updated Model**

```
<full reformulated LaTeX model>
```

**Important Guidelines**

- Prefer methods without new variables or binaries when possible

### A.2.3 MAX PATTERN PROMPT

**Role Definition**  You are an expert in reformulating **max** terms in mixed-integer optimization problems.

**Task Description**   For each $\max(\cdot)$ in the LaTeX optimization problem, do:

1. **IDENTIFY** – Quote exact max terms and its index set
2. **EVALUATE** – Check methods (**stop at first Applicable = YES, Exact = YES**):
   (a) Convex-hull
   (b) Indicator / Disjunctive (binary linking constraints)
   (c) Binary Big-M (derive tightest M explicitly)
   (d) Split-inequality (two simultaneous $\geq$ constraints) – **use only if a single argument of the min is proven to dominate globally; otherwise Applicable = NO**
   Summarize evaluation briefly (e.g., Split-ineq: Applicable=..., Exact=...)
3. **DERIVE M** – Clearly state M if used (else "n/a")
4. **FORMULATE** – Provide constraints in LaTeX; no other changes
5. **VERIFY** – Brief one-line justification

**Input Information**

- **LaTeX model:** {latex_model}
- **Pattern description (human hint):** {max_pattern}
- **Concrete parameters (bounds, indices):** {param_info}

**Output Format**

```
REPORT:
Pattern      : <max expression and indices>
Technique    : <chosen method>
Verification : <one concise sentence>
Bounds / M   : <bounds or Big-M; "n/a" if not used>
Aux vars     : <any new variables; minimal>
```

**Updated Model**

```
<full reformulated LaTeX model>
```

**Important Guidelines**

- Prefer methods without new variables

A.2.4   MIN PATTERN PROMPT

**Role Definition**   You are an expert in reformulating **min** terms in mixed-integer optimization problems.

**Task Description**   For each $\min(\cdot)$ in the LaTeX optimization problem, do:

1. **IDENTIFY** – Quote exact min terms and its index set
2. **EVALUATE** – Check methods (**stop at first Applicable = YES, Exact = YES**):
   (a) Convex-hull
   (b) Indicator / Disjunctive (binary linking constraints)
   (c) Binary Big-M (derive tightest M explicitly)
   (d) Split-inequality (two simultaneous $\geq$ constraints) – **use only if a single argument of the min is proven to dominate globally; otherwise Applicable = NO**
   Summarize evaluation briefly (e.g., Split-ineq: Applicable=..., Exact=...)
3. **DERIVE M** – Clearly state M if used (else "n/a")
4. **FORMULATE** – Provide constraints in LaTeX; no other changes
5. **VERIFY** – Brief one-line justification

**Input Information**

- **LATEX model:** {latex_model}
- **Pattern description (human hint):** {min_pattern}
- **Concrete parameters (bounds, indices):** {param_info}

**Output Format**

```
REPORT:
Pattern      : <min expression and indices>
Technique    : <chosen method>
Verification : <one concise sentence>
Bounds / M   : <bounds or Big-M; "n/a" if not used>
Aux vars     : <any new variables; minimal>
```

**Updated Model**

```
<full reformulated LaTeX model>
```

**Important Guidelines**

- Prefer methods without new variables

A.2.5    ABSOLUTE VALUE PATTERN PROMPT

**Role Definition**    You are an expert in reformulating **absolute value terms** in mixed-integer optimization problems.

**Task Description**    For each **absolute value term** in the LATEX optimization problem, complete steps A–E:

1. **IDENTIFY** – Quote exact absolute value terms and its index set
2. **EVALUATE** – Check methods (**stop at first Applicable=YES, Exact=YES**):
   (a) Split-inequality (two linear inequalities)
   (b) Convex-hull (linear inequalities with auxiliary variables)
   (c) Binary Big-M (**only if 1–2 fail**, explicitly derive tightest M)
   Summarize evaluation briefly (e.g., Split-ineq: Applicable=..., Exact=...)
3. **DERIVE M** – Clearly state M if used (else "n/a")
4. **FORMULATE** – Provide constraints in LATEX; no other changes
5. **VERIFY** – Brief one-line justification

**Input Information**

- **LATEX model:** {latex_model}
- **Pattern description (human hint):** {absolute_pattern}
- **Concrete parameters (bounds, indices):** {param_info}

**Output Format**

```
REPORT:
Pattern      : <absolute expression and indices>
Technique    : <chosen method>
Verification : <one concise sentence>
Bounds / M   : <bounds or Big-M; "n/a" if not used>
Aux vars     : <any new variables; minimal>
```

**Updated Model**

```
<full reformulated LaTeX model>
```

**Important Guidelines**

- Prefer simpler methods without binaries whenever possible
- Ensure mathematical equivalence

### A.2.6 QUOTIENT PATTERN PROMPT

**Role Definition**  You are an expert in reformulating **linear fractional terms (expressions of form** $y/x, x \neq 0$**)** into purely linear constraints for mixed-integer optimization problems.

**Task Description**  For each **linear fractional term** $y/x$ in the LaTeX optimization problem, complete steps A–D clearly and concisely:

1. **IDENTIFY** – Quote the exact linear quotient terms and its index set
2. **EVALUATE** – Check these methods (**stop at first Applicable=YES, Exact=YES**):
   (a) **Charnes-Cooper Transformation**
   (b) **Homogenisation / Normalisation**
   Briefly summarize evaluation (e.g., Direct: Applicable=..., Exact=...)
3. **FORMULATE** – Provide constraints explicitly in LaTeX; no other changes
4. **VERIFY** – One concise sentence explicitly confirming no nonlinearities or reciprocals remain

**Critical Guidelines**

- **DO NOT replace one linear fractional term with another**
- **DO NOT leave any nonlinear terms (e.g. bilinear terms) in the optimization problem where variables are involved**

**Input Information**

- **LaTeX model:** {latex_model}
- **Pattern description (human hint):** {quotient_pattern}
- **Concrete parameters (bounds, indices):** {param_info}

**Output Format**

```
REPORT:
Pattern      : <exact quotient expression and indices>
Technique    : <chosen method>
Verification : <one concise sentence explicitly stating linearity>
Bounds / M   : <bounds or Big-M; "n/a" if not used>
Aux vars     : <any new variables; minimal>
```

**Updated Model**

```
<full reformulated LaTeX model>
```

**Important Guidelines**

- Reformulated model must be purely linear (LP/MILP)
- **NO nonlinear terms or reciprocals remain**
- Prefer simplest methods without binaries if possible

### A.2.7 MONOTONE TRANSFORMATION PROMPT

**Role Definition**    You are an expert in reformulating **monotone transformations** in objective functions for mixed-integer optimization problems.

**Task Description**    Given an objective function of form $\min f(g(x))$ or $\max f(g(x))$, complete steps A–E clearly:

1. **IDENTIFY** – Clearly quote the original objective and explicitly identify:
   - Monotone function $f$ (e.g., $\log(x), x > 0$)
   - Linear function $g(x)$
2. **EVALUATE** – Confirm applicability (**all must be YES**):
   (a) Is $g(x)$ linear? (YES/NO)
   (b) Is $f$ monotone? (YES: increasing/decreasing, NO)
   (c) Is the transformation invertible? (YES/NO)
   (d) Is the domain well-defined? (YES/NO; state conditions explicitly)

   Provide a concise evaluation summary
3. **DERIVE TRANSFORMATION** – Explicitly state the equivalent linear reformulation
4. **FORMULATE** – Clearly provide the transformed (linear) objective in LaTeX, plus necessary domain constraints (e.g., positivity). **BUT** do not cut-off feasible points by this constraint
5. **VERIFY** – One concise sentence explicitly confirming equivalence and linearity

**Critical Guidelines**

- **ONLY apply if ALL criteria above are YES**
- Preserve optimization direction (min/max)
- Ensure NO nonlinear terms remain after transformation

**Input Information**

- **LaTeX model:** {latex_model}
- **Pattern description (human hint):** {monotone_pattern}
- **Concrete parameters (bounds, indices):** {param_info}

**Output Format**

```
REPORT:
Pattern : <exact original monotone objective>
Transformation : <clearly stated linear reformulation>
Verification : <concise sentence confirming linearity and equivalence>
Applicability : <YES/NO with brief reasoning>
Domain Conditions : <explicitly stated>
```

**Updated Model**

```
<full reformulated LaTeX model with linearized objective and required constraints>
```

**Post-Processing Step**    Clearly document how to recover the original objective value from the transformed solution.

**Important Guidelines**

- Ensure complete linearity after transformation
- Explicitly state all domain constraints required for correctness
- Prefer simplest transformations preserving optimization direction

### A.2.8 ONE-SHOT PROMPT

**Role Definition**     You are an Operations Research expert specialized in **exact linear reformulations** of nonlinear patterns in optimization models.

**Task Description**     From the given LaTeX model and parameter context:

1. Detect all linearizable nonlinear patterns.

2. Reformulate each **exactly** (MILP/LP) using the tightest valid technique.

3. Output a **single fully linear LaTeX model** that is mathematically equivalent.

**Input Information**

- **LaTeX model:** {latex_model}

- **Parameter context:** {parameter_context}

- **Parameter values:** {param_info}

- **Decision variables:** {sets_info}

**Patterns**     Please check for the following nonlinearity patterns and apply appropriate exact reformulation techniques:

- **Bilinear**

- **Min/Max**

- **Absolute**

- **Linear fractional**

- **Monotone transformation**

**Output Format     DETECTION**

```
- Nonlinearities: YES/NO
- Patterns: List specific patterns found, grouped by type
  - If no patterns found, write "NONE"
  - If patterns found, list them like:
    "Bilinear: x*y, Min: min(x,y), Max: max(x,y)"
```

**REFORMULATION REPORT**

```
For each detected pattern family:
- Pattern: expression and indices
- Technique: chosen method
- Bounds / M: numeric or "n/a"
- Aux vars: new variables
- Verification: one-line justification
- For monotone transformation: include the
  "POST-CODE HANDOVER (MONOTONE)" block
```

**REFORMULATED MODEL (LaTeX)**

```
- Full linearized model with:
  - Decision variables (including new ones)
  - Linear objective
  - Original constraints
  - New reformulation constraints
```

**REFORMULATION INFORMATION**

```
- Concise summary of transformations applied, bounds,
  Big-M values, and domain conditions.
```

**POST-CODE HANDOVER (MONOTONE)**

```
- Apply: YES/NO
- Direction: increasing/decreasing
- Targets: expressions/variables to transform (list)
- Bounds/domain: required bounds or domain conditions
- Transformation note: brief formula or mapping to
  apply after code generation
```

**FINAL CHECKS**

```
- Residual nonlinearities: NONE (or list if any)
- Big-M constants numeric and tightened: YES/NO
- Index sets consistent and defined: YES/NO
```

**Rules**

- Output must be LP/MILP only.

- Stop at first technique that is Applicable=YES and Exact=YES.

- Focus only on mathematical reformulation – code generation will be handled separately.

A.2.9    CODE CONVERSION PROMPT

**Role Definition**    You are an expert in Operations Research and Mixed-Integer Optimization. Convert the provided LaTeX-formulated optimization model into efficient and immediately executable Python code using Gurobi's gurobipy API.

**Task Requirements**

- Accurately translate the LaTeX model into gurobipy Python code

- Maintain complete mathematical equivalence between LaTeX and Python

- Clearly define all parameters, variables, constraints, and objectives

- Integrate the provided parameters explicitly and correctly

- Ensure the Python code is clear, efficient, and ready to run without modification

**Provided Information**

**Parameter Context**    {parameter_context}

**Available Parameters**    {param_info}

**Sets Information**    {sets_info}

**Reformulation Information**    {reformulation_context}

**LaTeX Model**    {latex_model}

**Guidelines for Code Generation**

- Start with imports: `from gurobipy import *`

- Define parameters with provided numeric values

- Map abstract indices and summations from LaTeX to concrete Python loops or efficient vectorized expressions

- Use gurobipy methods effectively:
  - `model.addVars()` for indexed variables
  - `model.addVar()` for single variables
  - `quicksum()` for efficient summations
- Clearly comment reformulations if applicable (e.g., "# McCormick envelopes for bilinear terms")
- Handle parameter dictionaries safely (e.g., `param.get(key, default)`)
- Finish your script with: `model.optimize()`

**Critical Quicksum Rules**

- NEVER use conditional expressions inside `quicksum()` like: `quicksum(x[i] for i in range(n) if condition)`
- Instead, use explicit loops with `LinExpr` or separate `quicksum` calls
- For conditional summations, use this pattern:

```
# WRONG: quicksum(x[i] for i in range(n) if condition[i])
# CORRECT:
expr = LinExpr()
for i in range(n):
    if condition[i]:
        expr += x[i]
```

Or use separate `quicksum` calls for different conditions.

**Important Guidelines**

- Do NOT include markdown formatting or fences
- Output should be only executable Python code with explanatory comments where necessary
- Generate ONLY executable Python code without any formatting, explanations, or markdown

## A.3 EXACT LINEARIZATION RECIPES

We collect compact examples showing how to derive exact LP/MILP counterparts for the nonlinear patterns used in the paper.

### A.3.1 BILINEAR PRODUCTS

**Case A: binary $\times$ binary.** Let $b_1, b_2 \in \{0,1\}$ and introduce $w = b_1 b_2$. An exact linearization is

$$w \leq b_1, \qquad w \leq b_2, \qquad w \geq b_1 + b_2 - 1, \qquad 0 \leq w \leq 1. \tag{1}$$

(Integrality of $w$ is implied when $b_1, b_2$ are binary.)

**Case B: binary $\times$ bounded continuous.** Let $b \in \{0,1\}$, $x \in [L, U]$ with known bounds, and $z = b\,x$. Then the following four inequalities are *exact*:

$$z \leq Ub, \qquad z \geq Lb, \qquad z \leq x - L(1-b), \qquad z \geq x - U(1-b). \tag{2}$$

When $b = 0$ these force $z = 0$; when $b = 1$ they force $z = x$.

### A.3.2 MINIMUM AND MAXIMUM OF LINEAR FUNCTIONS

**Constraint splitting for** $\min$. For linear functions $f_k(x)$, $k \in \mathcal{K}$, the constraint

$$t \leq \min_{k \in \mathcal{K}}\{f_k(x)\}$$

is equivalent to the set of linear constraints $t \leq f_k(x) \ \forall k \in \mathcal{K}$.

**Epigraph for** max **(objective).** To solve $\min_x \ \max_{k \in \mathcal{K}} f_k(x)$ introduce an auxiliary $z$ and use

$$\min_{x,z} \ z \quad \text{s.t.} \quad z \geq f_k(x) \ \forall k \in \mathcal{K}. \tag{3}$$

This epigraph encoding is the standard exact reformulation when max appears in the objective.

### A.3.3 ABSOLUTE VALUE

**Objective form.** To minimize $|t|$ (with $t$ linear), introduce $y$ and write

$$\min \ y \quad \text{s.t.} \quad y \geq t, \ y \geq -t. \tag{4}$$

**Equality form.** To enforce $y = |t|$ exactly without binaries, use the positive/negative parts:

$$t = t^+ - t^-, \qquad y = t^+ + t^-, \qquad t^+, t^- \geq 0. \tag{5}$$

### A.3.4 LINEAR FRACTIONAL (CHARNES–COOPER)

Consider the fractional objective

$$\min_x \ \frac{a^\top x + b}{c^\top x + d} \quad \text{s.t.} \quad Fx \leq g, \ c^\top x + d > 0.$$

Charnes–Cooper sets $t = \frac{1}{c^\top x + d} > 0$, $y = xt$, giving the LP

$$\min_{y,t} \ a^\top y + b\,t \tag{6}$$

$$\text{s.t.} \ Fy \leq g\,t, \qquad c^\top y + d\,t = 1, \qquad t \geq 0. \tag{7}$$

(Contrast with a *partial* cross-multiplication $u(c^\top x + d) \geq a^\top x + b$, which leaves the bilinear term $u\,x$ and is therefore nonlinear.)

### A.3.5 MONOTONE TRANSFORMATIONS

Let $\phi : \mathbb{R} \to \mathbb{R}$ be strictly monotone.

**Objectives.** If $\phi$ is strictly increasing, then $\min \ \phi(g(x))$ is equivalent to $\min \ g(x)$ (solve in $g$ and, if needed, report back the original value via $\phi$). Example: minimizing $\exp(s)$ is equivalent to minimizing $s$ (and reporting $\exp(s^\star)$).

If $\phi$ is strictly decreasing, then $\min \ \phi(g(x))$ is equivalent to $\max \ g(x)$.

**Constraints.** For increasing $\phi$, the constraint $\phi(g(x)) \leq \alpha$ is equivalent to $g(x) \leq \phi^{-1}(\alpha)$ (and similarly $\phi(g(x)) \geq \alpha \iff g(x) \geq \phi^{-1}(\alpha)$). Example: $\log(y) \leq \alpha \iff y \leq e^\alpha$.

**Notes.** The bilinear case with one binary and one bounded continuous variable is exact with the four McCormick inequalities above; if both variables are continuous, McCormick gives a relaxation. For fractional objectives, positivity of $c^\top x + d$ on the feasible set is required.

### A.4 TECHNICAL RESULTS GENERATION METHODOLOGY

This section describes the experimental methodology for evaluating the *LinearizeLLM* framework with OpenAI's *o3* (`temperature`$=0.0$[1], `top-p`$=0.9$, `max tokens`$=16000$, `timeout`$=120$ s) and Google's *Gemini 2.5 Flash* (`temperature`$=0.15$, `max tokens`$=16000$, `timeout`$=120$ s) across 20 optimization problems and three information scenarios.

**Experimental Framework.** The *LinearizeLLM* framework processes LaTeX optimization problems through pattern detection, MIP linearization, and code generation. We evaluate three information scenarios: *no_context* (LaTeX only), *partial_info* (with decision variable types, no parameter information), and *full_info* (complete parameter information). Generated code is executed using *gurobipy* with default settings.

---

[1]Blend #1, Blend #2, and Network #1 were run with `temperature`$=0.10$; Diet #3 and Network #2 with 0.15.

**Reproducible Experimental Design.** Each experiment uses five fixed seeds (1, 2, 3, 4, 5) to ensure reproducibility, generating 300 runs per model (20 problems × 3 scenarios × 5 seeds). All experiments were conducted on a system with AMD Ryzen Threadripper PRO 7955WX processor @ 4.5 GHz, 512 GB RAM, and PNY NVIDIA T1000 8 GB graphics card. Detailed instructions for executing these experiments can be found in the project `README.md`.

**Performance Evaluation.** Results are evaluated against ground truth solutions using success rates (OSR, DSR, RSR, CSR), where we solve the most tractable form of each nonlinear problem using *gurobipy* with default parameters and compare against our reformulated solutions generated by *LinearizeLLM*.

