# OpenReview forum: "LinearizeLLM: An Agent-Based Framework for LLM-Driven Exact Linear Reformulation of Nonlinear Optimization Problems"
_ICLR.cc/2026/Conference — ICLR 2026 Conference Desk Rejected Submission_

### Official Review · Reviewer_XAr7 · 2025-10-19

**Soundness:** 2
**Presentation:** 3
**Contribution:** 2
**Rating:** 2
**Confidence:** 5

**Summary:**

LinearizeLLM is a framework that uses large language models (LLMs) to automatically detect and reformulate nonlinear optimization problems into equivalent linear programming (LP) or mixed-integer linear programming (MILP) formulations.
It transforms nonlinear mathematical expressions into solver-ready linear models by combining multi-agent reasoning, pattern recognition, and structured prompting.

**Strengths:**

1. The paper explores an interesting intersection between large language models and mathematical optimization, showing how reasoning-based LLMs can be applied to structured, symbolic problems.
2. The modular multi-agent design is conceptually clear and demonstrates that LLMs can perform pattern detection and rule-based transformations in a coordinated workflow.

**Weaknesses:**

###  Contrived Problem Formulation
The core task—detecting and reformulating nonlinear expressions into linear or MILP equivalents—is not a real bottleneck in mathematical optimization practice.  Existing modeling languages (e.g., Pyomo, JuMP, AMPL) and symbolic engines (e.g., SymPy) already handle these transformations deterministically and provably correctly.  Even if the goal is to teach non-experts how to formulate optimization models, I don't think LLM is the right tool for the task. Using a symbolic backend to give the user exact feedback is much better than using the LLM.

###  Limited Novelty in Methodology
The paper’s core contribution, using multiple LLM “agents” for detection and reformulation, represents an engineering orchestration, not a new algorithmic insight.  Existing literature on optimization and LLM has already used similar agentic frameworks.

### Evaluation Weaknesses
The experiments rely on small, synthetic datasets (20 problems, many artificially injected nonlinearities).
No results are reported on real optimization benchmarks or solver runtime impacts, making it unclear whether the system produces usable models in practice.

### Overstated Claims of Generalization
The paper implies broad applicability across nonlinear structures, yet all examples belong to a limited set of hand-crafted patterns ( abs, min/max, fractional).

## Potential improvements
To make is acceptable for publication, it might be of interest to consider problems where there no existing methods in the literature to linearize. The LLM can potentially discover novel ways to reformulation the MINLP.

**Questions:**

1. How does the system ensure correctness beyond heuristic validation? For example, is there any symbolic or numerical verification that the reformulated linear model preserves feasibility and optimality?
2. Given that existing modeling frameworks already automate many of these transformations, what unique scenarios or input formats justify the use of an LLM-based approach instead of traditional symbolic parsing?

---

### Official Review · Reviewer_nhht · 2025-10-27

**Soundness:** 2
**Presentation:** 2
**Contribution:** 3
**Rating:** 4
**Confidence:** 4

**Summary:**

The paper proposes LinearizeLLM, a multi-agent pipeline that takes a LaTeX description of a nonlinear optimization problem and iteratively detects specific nonlinearity patterns, then replaces each with an exact linear reformulation. The workflow has three stages: pattern detection, pattern-specific reformulation and iteration until no target nonlinearities remain. Experiments are conducted on 20 ComplexOR instances by injecting linearizable patterns and report different level success rates, comparing Gemini 2.5 Flash vs OpenAI o3 under the proposed agentic loop. The staged approach generally improves overall success rate over baselines.

**Strengths:**

- Reformulation of linearized optimization problem to LP or MILP is new and rarely studied systematically using LLMs. Pushing LLMs beyond "text→model" into model improvement is a meaningful step for OR+LLM.
- The detect, reformulate loop is a clear, modular workflow and agent decomposition are easy to understand and plausibly extensible to more patterns.
- The paper covers textbook transformations (epigraph for max, absolute value linearizations, Charnes–Cooper for linear fractional, big-M for binary–continuous interactions, monotone transforms with inverse for reporting), with decent diagnostics of failure modes.

**Weaknesses:**

- Limited framework novelty: the proposed detect → reformulate → iterate loop resembles a conventional rule-driven rewrite pipeline already common in LLM agent systems, rewrapping standard linearization rules behind prompts without introducing new mechanisms or correctness guarantees.
- Narrow evaluation: the benchmark consists of only 20 ComplexOR instances with injected patterns, which severely limits external validity and risks distributional overfitting to clean, non-nested patterns.
- Baselines are limited: comparisons are only between LLMs and a one-shot prompt; deterministic rule-based rewriters (e.g., Pyomo, AMPL/CVX/YALMIP transforms) are strong baselines and should be included, along with a discussion on how the LLM-based reformulation relates to these rule-based methods.
- Metrics lack absolute transparency: Table 2 presents relative values against baselines without absolute numbers per pattern and instance (and runtime), making it difficult to assess real-world reliability or overhead.

**Questions:**

- Can machine-checked guarantees be attached for each reformulation step—for example, by generating proofs that a theorem prover (e.g., Lean) can discharge or by emitting certificates that can be verified offline? Could the LLM propose the reduction and proof sketch while a verifier certifies it before applying? See Bentkamp et al. (2023).
- Can you add real, messy examples (e.g., from MINLPLib) and include nested patterns such as $\max(|a^T x|, g(x))$? Please report the performance of the pipeline, including where it fails and why, and provide absolute per-pattern and per-instance scores and end-to-end runtimes for each baseline.
- Please add deterministic, rule-based baselines (e.g., Pyomo with generalized disjunctive programming) and explain where your LLM method improves on them. A thorough discussion on the differences between AI-agent and deterministic methods would strengthen the conclusion.

---

### Official Review · Reviewer_kNgt · 2025-10-28

**Soundness:** 3
**Presentation:** 3
**Contribution:** 3
**Rating:** 6
**Confidence:** 3

**Summary:**

This paper presents LinearizeLLM, an agent-based LLM framework that automatically converts nonlinear optimization problems (NLPs) into algebraically equivalent linear (LP/MILP) forms.
Each nonlinear pattern (e.g., bilinear terms, |·|, min/max, linear fractional, monotone transformations) is handled by a dedicated reformulation agent that applies exact linearization techniques. The framework iteratively detects, reformulates, and verifies nonlinear patterns until a solver-ready linear model is obtained.
A new benchmark of 20 real-world nonlinear problems (derived from ComplexOR) is released to evaluate the method. Experiments with Gemini 2.5 Flash and OpenAI o3 show that LinearizeLLM significantly outperforms one-shot and non-agent baselines—achieving up to 107% improvement in overall success rate (OSR) and near-perfect detection/reformulation success on several nonlinear patterns.

**Strengths:**

* Innovative agentic decomposition for algebraic reformulation—extends prior “Chain-of-Experts” frameworks toward symbolic linearization.
* Transparent and auditable pipeline, improving explainability compared with solver black boxes.
* Empirical evidence supports consistent performance improvement, especially on complex nonlinearities.
* Potentially impactful for democratizing optimization modeling—bridging LLM reasoning and mathematical tractability.

**Weaknesses:**

* Limited scale and diversity of experiments (20 synthetic instances).
* No formal validation of algebraic equivalence beyond numerical consistency.
* Dependence on specific prompt templates—unclear generalization to unseen nonlinearity structures.
* No runtime or efficiency analysis—LLM cost, inference time, or token usage are missing.
* Baseline scope (only one-shot and another model) is insufficient to prove general superiority.
* Illustration of the LinearizeLLM workflow in figure 1 is quite confusing. Further explanation in caption will be helpful.

**Questions:**

* What is the computational overhead (LLM calls, tokens) compared to manual reformulation or solver preprocessing?
* How sensitive is performance to prompt phrasing or context length? The context-blind ablation hints at brittleness—can this be mitigated via meta-reasoning?
* How does the system ensure algebraic equivalence beyond solver outcome matching? Could symbolic simplification or constraint hashing be used for formal verification?

---

### Official Review · Reviewer_ccd4 · 2025-10-31

**Soundness:** 2
**Presentation:** 2
**Contribution:** 2
**Rating:** 2
**Confidence:** 3

**Summary:**

This paper introduces **LinearizeLLM**, an agent-based framework that automates the exact linear reformulation of nonlinear optimization problems using Large Language Models. It addresses the expertise gap in operations research by employing specialized reformulation agents, each trained to detect and linearize specific nonlinear patterns—such as absolute values, bilinear terms, and fractional expressions—into equivalent linear or mixed-integer linear models. The system iteratively processes a problem until it becomes solver-ready. Evaluated on a benchmark of 20 real-world problems derived from ComplexOR, LinearizeLLM with Gemini 2.5 Flash significantly outperforms one-shot and other LLM baselines, achieving high success rates in detection, reformulation, and overall model correctness. This work demonstrates the potential of LLM-driven agents to make advanced optimization techniques accessible to non-experts.

**Strengths:**

1.  **Effective Multi-Agent Specialization:** The framework's core strength is its decomposition of the complex linearization task into specialized agents, each an expert in a specific nonlinear pattern (e.g., bilinear, absolute value). This targeted approach, guided by structured reasoning checklists, proves significantly more reliable than a single, one-shot LLM prompt, leading to higher success rates and more robust reformulations.

2.  **Enhanced Auditability and Portability:** Unlike opaque solver-internal reformulations, LinearizeLLM produces a transparent, human-readable linear model. This allows users to verify the correctness of the introduced auxiliary variables and constraints, building trust. Furthermore, the output is a standard LP/MILP that can be ported to any compatible solver, ensuring flexibility and independence from proprietary nonlinear solvers.

**Weaknesses:**

1.  **Limited Scope and Insufficient Experimental Evidence:** The experimental benchmark is narrow, comprising only 20 custom-made problems derived by injecting specific nonlinearities into an existing linear dataset. This small, synthetic test set is insufficient to robustly support the paper's claim of handling a "broad class" of real-world nonlinear problems. A more convincing evaluation would involve a larger, diverse set of native nonlinear problems from established MINLP libraries.

2.  **Marginal Novelty in Core Concept:** The fundamental concept of using pattern-specific rules for linearization is a well-established, textbook practice in operations research. The paper's primary novelty lies in orchestrating this process with LLM agents rather than hard-coded rules. While the LLM-based automation is a contribution, the underlying linearization techniques themselves are not new, and the agent-based workflow is an incremental step over existing LLM frameworks like Chain-of-Experts.

3.  **Dependence on Predefined Patterns and Assumptions:** The framework's effectiveness is confined to a predefined set of six, mutually independent nonlinear patterns. It cannot handle nested nonlinearities or patterns outside its trained agents' knowledge. This lack of generalizability is a significant limitation, as real-world optimization models often contain complex, interdependent nonlinear relationships that fall outside these clean, pre-defined categories.

**Questions:**

same as the weakness:

* **Limited Scope and Insufficient Experimental Evidence**
* **Lack of Novelty**
* **Dependence on Predefined Patterns and Assumptions**

I will be happy if the authors will solve my issues, then I would like to raise my score.

---

### Note · Program_Chairs · 2026-01-17
**Submission Desk Rejected by Program Chairs**

The following references in this submission do not refer to real documents and/or have major errors in bibliographic information:

 Qi Huangfu, J. Hall, Julian A. Antonio Frangioni, Ted K. Ralphs, and Matteo Galati. HiGHS: A high-performance open-source solver for large-scale linear optimization. SoftwareX, 23:101451, 2023. doi: 10.1016/j.softx.2023.101451. URL https://doi.org/10.1016/j.softx. 2023.10145